# A Measure of Tourist Responsibility

**Álvaro Dias** [1] , **Inês Aldana** [2] , **Leandro Pereira** [3,*] , **Renato Lopes da Costa** [3] and **Nelson António** [3]

1   Universidade Lusófona/TRIE, 1749-024 Lisbon, Portugal; alvaro.dias@ulusofona.pt
2   DMOGG, ISCTE-IUL, 1649-026 Lisbon, Portugal; ines.aires@hotmail.com
3   BRU-Business Research Unit, DMOGG, ISCTE-IUL, 1649-026 Lisbon, Portugal;
    Renato_Jorge_Costa@iscte-iul.pt (R.L.d.C.); nelson.antonio@iscte-iul.pt (N.A.)
*   Correspondence: leandro.pereira@iscte-iul.pt

**Abstract:** In a post-pandemic context, destinations are questioning mass tourism, and are focusing on more sustainable segments, looking for more responsible tourists. This requires obtaining relevant information to assess what kind of tourists visit the destination and, at the same time, to monitor changes in tourists' behavior and attitudes. This study aims to respond to this challenge by creating a measure to assess the tourist's responsibility. Using a scale development method, a sequential mixed-method approach is conducted to identify scale dimensions and items. An initial qualitative approach is implemented for item generation using focus group and face-to-face interviews. Then, a second study based on a survey is conducted for exploratory factor analysis. A third study, also based on a survey is performed to obtain a new sample for confirmatory factor analysis. Findings show two dimensions: civic responsibility and philanthropic responsibility, allowing an understanding of how tourists can act responsibly in destinations without compromising the ecological footprint on the planet. Theoretical and managerial implications are discussed.

**Keywords:** responsible tourism; tourist responsibility; scale development; sustainability; PLS

## 1. Introduction

Tourism has grown at an unrestrained rate all over the world. The desire to travel increases with the need to escape routine and the desire of the tourist to get to know new places and cultures. Tourism is an industry that moves millions of people, from travelers to employees working directly or indirectly [1]. However, there is growing concern about the negative impact of tourism and sustainability, with more attention being paid to the tourists' responsibility [2]. Furthermore, tourism responsibility has been playing an increasing role in the destination competitiveness [3] and sustainability. Chettiparamb and Kokkranikal [4] argue that responsible tourism initiatives is considered a destination tourism management strategy, covering planning, product development and marketing management to generate positive economic, social, cultural, and environmental impacts.

Despite the importance of measuring and monitoring the tourist responsible behavior, to our best knowledge, there is no scale to measure the concept of responsibility from the tourist perspective elaborated based on scale development methodology. Although Blackstock et al. [5] propose a measure, they recognize the limitation of their qualitative approach by considering "the sample is not representative of any total population" (p. 282). The Gao et al. [6] scale explores the link between tourists' perceptions of the negative impacts of tourism and the perception of their responsibility in the territories. The authors also use exploratory and confirmatory research methods and questionnaires to present the results. Their proposal focuses mainly on tourists' perceptions of responsibility and the perceived negative impacts of tourism, lacking a real understanding of the notion of tourist responsibility at the destinations. To address this gap, this study aims to propose a measure of tourist responsibility that can contribute to both academics and practitioners to monitor and compare the level of responsibility of the tourists visiting destinations. This article also

aims to contribute for more sustainable destinations by providing a tool to measure the visitors' attitude towards more responsible behaviors.

The article is structured as follows. Nest section is dedicated to the theoretical framework, presenting the key concepts, a discussion about tourist responsibility and revealing the potential issues for the scale development. Section 3 details the methodology, more specifically how will be conducted the scale development and the respective steps. Section 4 presents the results of the sequential steps for scale development. Finally, the conclusions detail the theoretical and managerial implications as well the limitations and proposals for future research.

## 2. Theoretical Framework

### 2.1. Key Concepts

There are several authors who deal with the definition of responsible tourism. Responsible tourism began to gain notoriety during the 1980s, associated with the emerging concept of tourism sustainability [7–9], and which today is treated as an imperative in the strategic management of tourism in destinations. Wheller [10] already witnessed the changes in the last 20 years regarding the attitudes of consumers and suppliers, which emphasized the need for sustainable tourism development, namely to minimize the negative impacts of tourism on the environment. Poon [11] approach defends that the construction of responsible tourism is fundamental for the dignified and planned development of tourist destinations, in particular for the controlled management of the destination's capacity.

For those who practice it, responsible tourism is seen as an alternative tourism practice to mass tourism, appropriate, conscious, light and green, and that does not have any negative repercussions on the host environment [8]. Responsible tourism can be considered as a positive alternative to mass tourism and capable of replacing it [12], but it can also play a significant role on mass tourism itself [13]. They defend the position of the South African Department of Environmental Tourism Affairs that it is a type of tourism capable of promoting responsibility in the sustainable use of the environment; capable of involving local communities in tourism activity; and capable of responsibly ensuring the safety of visitors and all those involved. In this way, it can be seen that the application of responsible tourism is beneficial for the controlled and sustainable development of a territory. Of interest to all parties involved, its practice in destinations is important for sustainable tourism development and management, capable of properly managing resources in territories and contributing to increasing the quality of life of local communities. Responsible tourism maximizes benefits for local communities and minimizes negative social and environmental impacts, helping people to conserve their cultures and habitats.

Ethical values are fundamental to the good practice of these environmental and social principles in destinations [14]. The foundation of responsible tourism in territories is applied through sustainable tourism initiatives that make it possible to build a better place for communities to live and create better business opportunities for tourism companies, which, in turn, generate better experiences for the tourists they visit [12]. Responsible tourism is supported by basic principles such as respect for others, the environment and responsible actions of each one. It is based on appropriate strategies and policies supported by sustainability, aggregated with appropriate behavior and capable of performing (re)sustainable actions, responsive to and sustained by environmental and ethical tourism awareness [14–16]. Goodwin [17] also confirms that the terms of sustainable tourism and responsible tourism in the literature are often mixed, but they cannot be treated in the same way. Responsible tourism portrays how to deal with sustainability issues. The several perspectives are presented in Table 1.

**Table 1.** Responsible Tourism Perspectives.

| Perspective | Authors |
|---|---|
| Association with Tourism Sustainability | Camilleri [7]; Caruana et al. [8]; Goodwin [17] |
| Alternative and conscious tourism practice | Caruana et al. [8,16]; Wheller [10]; Chan and Xin [12] |
| Development and sustainable tourism management in the territories | Wheller [10]; Chan and Xin [12]; Leslie [15] |
| Resource management and improving the quality of life of communities | Chan and Xin [12]; Mathew and Sreejesh [18] |

Tourist responsibility thus falls to all stakeholders in the tourism chain, as depicted in Figure 1. Responsible tourism implies that everyone involved in tourism is responsible for the consequences of their behavior: tourism companies, institutions and local communities, destination managers (Destination Management Organization (DMO)), investors, consumers, etc. [19–21].

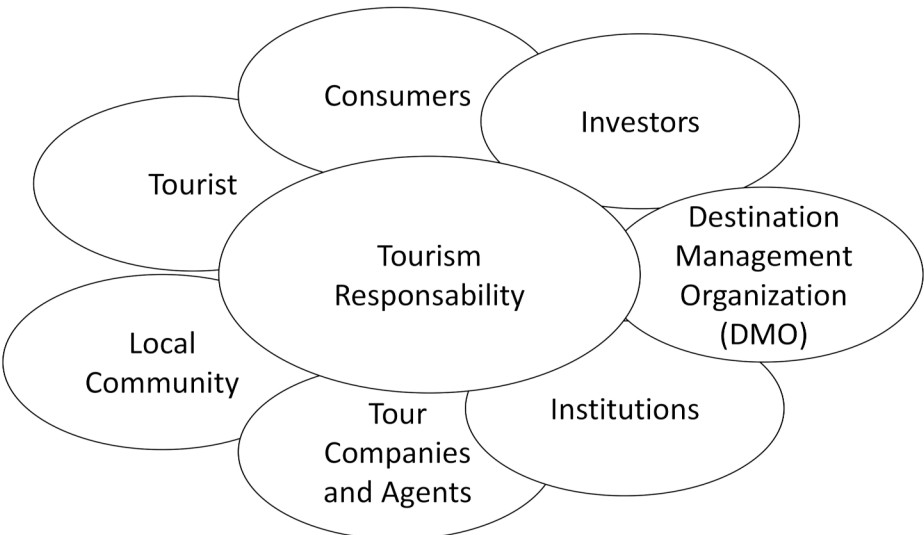

**Figure 1.** Stakeholders influence tourism responsibility.

Highlighting the perspectives of responsibility from the tourist point of view, Wheller [10] states that the awareness of the traveler before arriving at the destination is fundamental, and on the basis of this understanding is his education. The responsibility of the tourist passes through the conscious and responsible thought, both environmental and ethical [16]. For Krippendorf [22], responsible tourists are willing to invest adequate time and resources, and seek information before traveling to destinations in order to live local experiences consciously and ethically.

Governmental entities and tourism agents, on the other hand, consider tourism responsibility essential to the sustainable development of destinations. The United Nations World Tourism Organization UNWTO [23], in 1999, projected the guiding principles for the participants in tourism (governments, tourism industry and communities), through the realization of the global ethical code for tourism, which aims to help maximize the benefits of the activity, minimizing the negative impact on territories, cultural heritage, and local communities around the world. In 2005, it carried out a tourist ethical code for the practice of responsible tourism activity. The brochure on responsible tourism approved by the World Committee on Tourism Ethics (WCTE) [24], is the perfect example that responsible tourism is not only the responsibility of the managers of the activity, but also of the tourist himself who must commit himself to contribute to a conscious activity and demonstrate sensitivity to multidimensional issues involving tourism, showing, in turn, as a quality visitor or guest. This ethical code of tourism responsibility shows that tourism must be

driven by education, tolerance and learning about the legitimate differences that exist between peoples and their cultures [25].

Nevertheless, the importance of the local community, which is also responsible for ensuring the conscious practice of tourism in the territories [26]. The involvement and participation of local communities with the entities responsible for tourism management is fundamental for the establishment of balanced negotiations between stakeholders, as well as for controlling the use of common resources that are used for tourism [27].

*2.2. Current Research on Responsible Tourism*

Loda and Macrì [28] argue that the concept of "responsible tourism" has been spreading rapidly in recent years. Responsible tourism evokes the conscious behavior of the tourist in order to minimize negative impacts on destinations, in environmental, economic, and cultural contexts. Its research tries to define the concept within the development of a sustainable plan for Myanmar cities by creating an index to measure the responsible behavior of tourists on site.

Booyens and Rogerson [29] state that responsible tourism incorporates economic, environmental, and social dimensions. They point to the innovation of tourism entities in environmental and socially responsible practices. The article proposes a study of innovation tourism in the Western Cape of South Africa on responsible tourism. It provides a framework for conceptualizing and cultivating tourism innovation based on sustainability. The results point to the broad implementation of environmental innovations and practices by tourism enterprises. The authors also point out that local policy initiatives are needed to strengthen innovation for responsible business practices in tourism and thus enable environmental and social change on a larger scale.

Klein's [30] study on cruise tourism is concerned with the three areas focused on responsible tourism: the environment, the economy, and socio-cultural conditions, and considers strategies to ensure the sustainable development of cruise tourism. The research allowed an analysis of the perceptions of host communities and their concerns. The results showed that the sustainable growth of cruise tourism is possible through responsible tourism.

Additionally associated with the sustainable concern of the tourist territories, it has been an emerging subject in several studies related to tourism sustainability, as it is presented as a conscious practice for the tourist to carry out the activity. This is the case of the study by Sharpley [31], which shows that in recent years tourists have shown their need to experience new tourist practices, different from everyday life. Voluntary tourism is referred to as the ally of tourism, responsible for the set of conscious recommendations that it presents about this type of trips, both for tourist service providers and travelers. Sharpley [31] mentions that, at the heart of the practice of responsible tourism is a gentle and careful impact on the tourist and the stakeholders, on the cultural and natural environment of the destination and on local communities. The focus is on different types of responsible tourism: social, ecological, religious, and fair. Chan and Xin [12] noticed that responsible tourism is applied on the basis of the principle of sustainability. The authors point out the definitions, practices, and challenges of responsible tourism through the application of interviews with 25 tour operators and park managers, as well as through research and data analysis. Their results indicated that responsibility was built, and therefore possible to be applied, around the principles of sustainability, ecology, economic viability, and cultural sociability in Kinabalu Park, Sabah.

When it is admitted that there is a range of types of tourism considered responsible tourism, volunteer tourism is among the first to be mentioned. One example is ecotourism. Choi et al. [32] introduce transformation plans that stimulate ecotourism in Korea, proposing plans to improve responsibility in tourism sites: in the first plan, local residents should develop a system to manage, operationalize, and cooperatively govern ecotourism enterprises; in the second plan, ecotourism operators should improve the quality of educational programs of tourism information platforms in order to raise awareness of ecotourists' re-

sponsibility; and in the third plan, ecotourism systems should be improved by ecotourists and tour operators through a sense of responsibility. This study is significant in that it discusses the role of stakeholders in ecotourism planning and the promotion of responsible tourism and its role in the use and conservation of natural resources.

Other studies related to consumer perceptions, attitudes and behaviors towards responsible tourism show discrepancies between the attitudes that require the tourist to practice the activity responsibly, compared to their actual behavior [33]. Investigations by other authors reveal that few tourists consider themselves willing to change their behavior effectively to transcend more sustainable behavior [33]. For Bramwell et al. [34] and Budeanu [33], it is difficult to define the main barriers for the tourist to choose to be responsible, i.e., there is some difficulty in understanding what the motivations of a responsible tourist are and how their practices are measured [35]. For Del Chiappa et al. [2], in their study on responsible tourism, they found that it is an increasingly observed phenomenon. They decided to carry out an investigation on the factors that prevent tourists from traveling responsibly. A sample of 837 Italian travelers was carried out, describing the main impediments to the practice of responsible tourism. The results of the study suggest that the impediments to responsible tourism are related to five main categories: "lack of accessibility", "lack of will", "lack of reliability", "stress", and "price".

Responsible tourism practices were identified as an ideal framework to sustain the optimal growth of tourism and minimize the negative impacts on its development [25,36]. Their research examines the moderating effect of responsible tourism practices on the relationship between tourism development and quality of life. They showed that residents admitted that responsible tourism practices positively affect their quality of life.

Still in the field of tourism responsibility, it is possible to verify more recent studies related to the responsible behavior of the tourist and his influence on the environment. In the research of Wang et al. [37], they mention that in tourism activities the responsible environmental behavior of tourists is the result of positive human interaction with the environment. Their research uses the theory of planned behavior, carried out in the case study on Huangshan Mountain, China. Based on 534 questionnaires of tourist samples, the results show that tourists' intentions regarding environmental behaviors positively affect their responsible environmental attitudes. Another example is that of Said [38], motivated by tourism in Egypt, wanted to know if the tourist acts responsibly in Siwa. The author mentions that Siwa has always been considered an ecotourism destination due to the uniqueness of its cultural attractions and the fragility of its diverse ecosystems. Despite the rich literature about Siwa, its attractions, the sustainable development of tourism and the conservation of its assets, few studies exist on a responsible approach to tourist behavior. Tourists going to ecotourism destinations must have high environmental awareness and responsible behavior. While tourists go to ecological areas because they are attracted by natural resources, not everyone gets involved in positive environmental behavior. This study aimed to investigate the environmentally responsible behavior of tourists in Siwa using a quantitative approach. The author developed a questionnaire composed of 21 attributes of responsible tourism behavior before, during, and after the trip and delivered it in turn to tourists on site. The measurement of the behavior was done on a three-point scale (always, rarely, never). The questionnaires were applied to tourists in Siwa during the period from January to May 2018. The results showed a moderate behavior of the responsible tourist in Siwa.

For Bulin [39], the concept of responsibility is also related to aspects such as ethics, social responsibility, sustainable development, and sustainable tourism, offering various research options. The objective of his article is to define responsible tourism and to characterize the responsible tourist, through statistical analysis and his opinion on responsibility in tourism. The study is based on quantitative research—a questionnaire applied to Romanian tourists. The study by Lee et al. [40] explores behavioral discrepancies in responsible tourism practices at three ethical levels: economic, socio-cultural, and environmental. In

the results the authors present that tourists act on specific issues of responsibility differently during trips.

The reality of studies in the area of social, ethical, and political responsibility, from the perspective of the tourism stakeholders, is also evident, but it is also important for the increase in tourism responsibility in the destination. Su et al. [41], have developed an integrated model to demonstrate how the social responsibility of a destination influences the impacts of tourism (positive and negative impacts), the general satisfaction of the community, and how it directly and indirectly influences the environmentally responsible behavior of residents in the territories. The model was examined from a sample of 453 residents living on Gulangyu Island, a famous tourist destination on Xiamen Island in China. The results show that the destination's social responsibility increases residents' perception of the positive impacts of tourism, improves overall community satisfaction and contributes to the residents' responsible and environmental behavior.

## 3. Methodology

Scale development has been carried out in recent times in various studies to evaluate attitudes, techniques, and other scientific applications. The construction of a measurement scale offers the possibility to researchers of knowledge about people, concepts, and other processes. The development of a scale becomes an essential tool for measuring phenomena that before were not possible to be measured. The phenomena are items created by theoretical variables not observable in a direct environment [42].

In different investigations, it can be seen that the construction of a measuring scale is carried out using a step procedure. The following three basic steps are frequent: the generation of items; the purification of the generated items; and the evaluation of the reliability and reliability of the measurement scale construction [42–44]. For DeVellis [42], after generating the items, the purity of the generated items is evaluated, which ensures that to test the scale validation it is necessary the evaluation of other experts, usually using expert judgment as a method, and also when testing through the target audience to be reached, being possible through the elaboration and evaluation of questionnaires. For the authors, the third and last step refers to psychometric analysis. The researcher must test if the scale has validation and construction reliability. Churchill [45] states that this is the fundamental step to know if the content of the study is really measured. It is at this stage that specialized programs for statistical analysis are introduced, such as IBM SPSS and SmartPLS 3, which are used in this study. Through IBM SPSS the validity construction is ensured through the exploratory factor analysis, called Exploratory factor analysis (EFA), and through SmartPLS 3, the confirmatory factor analysis, (CFA)—Confirmatory factor analysis. Reliability is measured with the consistency of each item score. These tests are guaranteed by representative samples [42].

Other studies involve more than three essential steps for the development of a measuring scale. This is the case, among studies, of the research of Thomas et al. [46], in an essay on the development of a scale for the involvement of the wine vintner in wine destinations; present the development of its scale through seven stages. For the researchers, the first stage is the construction of the definition and design of the scale, which is justified by the literature review; the second stage, the generation of items, provided by qualitative studies, such as the literature review and methods such as the focus group; the third stage refers to the expert judgment that validates the content; the fourth stage, the purification of the items through the first samples applied by questionnaires; the fourth stage refers to the purification of the items through the first samples applied by questionnaires; the fifth stage refers to the initial validation, where the exploratory factor analysis is applied and the reliability of the generated items is tested; the sixth stage refers to the final validation that is conferred by new samples that are evaluated by confirmatory factor analysis and discriminant and nomological validations are performed; the seventh and last stage refers to the evaluation of the final construction of the tourist's scale of measurement in wine destinations compressed to seven attributes (see Figure 2). Similar to this last demonstra-

tion of scale, but based on the procedure of the development of the scale of Tsaur et al. [47] on the tourist-resident conflict, the following flowchart is adapted and presented for the resulting investigation composed of four main stages:

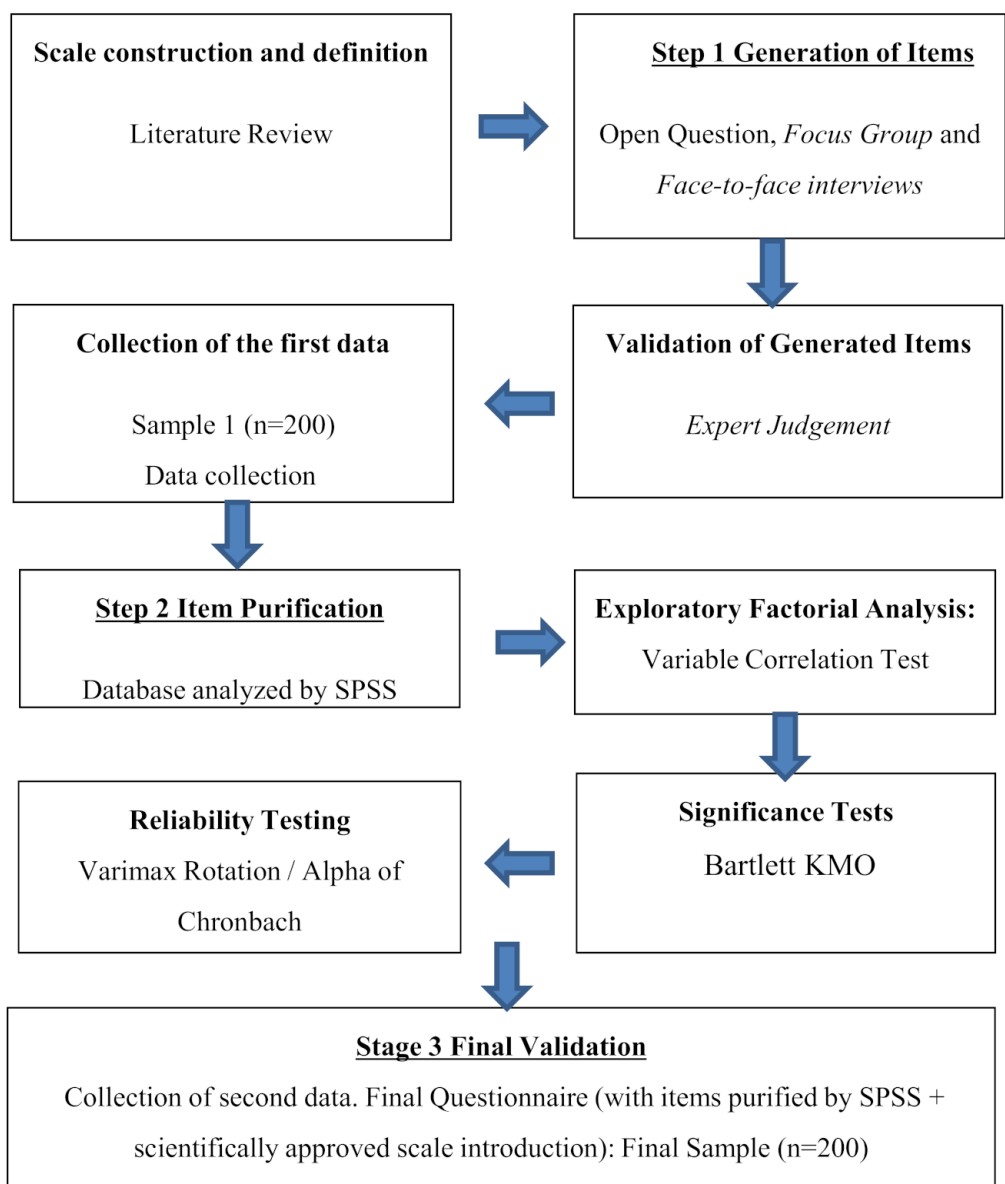

**Figure 2.** Flowchart of the procedure for the development of the Tourist Responsibility measurement scale from the tourist perspective.

When analyzing the background of science and the most recent research, it is clear that the conscious and sensitive competence of tourism stakeholders is fundamental, and how the social, ethical, cultural, and environmental dimensions are intrinsically related to the practice of responsible tourism. In order to be a 'responsible tourist', the tourist needs to take into account the different multidisciplinary notions of tourism. By understanding the dimensions of tourism responsibility it will be possible to create a scale of measurement. To measure, six studies are proposed: two qualitative and four quantitative. The results are described in the next section.

## 4. Results

### *4.1. Step 1: Item Generation*

The first stage is the generation of items reflected through prior research of the literature review on the theme, as well as the use of qualitative studies such as face-to-face or group interviews [48].

Qualitative Studies

In the qualitative studies, a first study of open questions is made: one positive and one negative, so that respondents have the opportunity to expand their ideas. The following questions are asked: 'What is it to you to be a responsible tourist?' and 'What do you understand by tourist irresponsibility?' In the second qualitative study, an Expert Judgment is conducted to evaluate the items generated in the first qualitative study.

Open Question

In order to not restrict the critical thinking of the respondents with regard to their opinion on responsible tourism, an open question was established for comment. The advantages of the open-ended question study allow respondents to give a more comprehensive opinion, enabling all ideas; it also allows them to express themselves in their own words, which makes the answers more impersonal, more comfortable, and also makes it possible to make distinctions, which are generally not possible in closed-ended questions [49]. The fact that there are two questions, one positive and one negative, also makes it easier to answer.

Respondents should respond on the basis of their knowledge of the subject and be forthright in stating their opinion. This subjective experimental method assumes the objective of formulating different possible independent variables that may be related to the dependent variable: tourism responsibility. By explaining the phenomenon of responsibility in tourism, the respondent will help to establish causal relationships and to determine the dimensions, which will contribute to the measurement of tourist responsibility.

Methods Used for the Development of the Instruments: Focus Group and Face-to-Face Interviews

A total of two methods were used for the development of the instruments: the Focus group and Face-to-face interviews.

A Focus group is a qualitative data collection technique, with the aim of obtaining group responses. This method is effective because through the response of the group members it is possible to extract feelings and opinions that constitute a new knowledge [50]. Face-to-face interview data collection is another form of interpersonal communication. It is an effective method in that communication between those involved is not only verbal but also nonverbal. Through body attitude, gestures, tone of voice, and facial expressions it is possible to identify the behavior of the respondent and thus obtain the transmission of the message he or she wants to convey [49]. The development of the instruments is based on the scale development process used by several researchers [37,42,45]. The instruments used in the research are questionnaires.

In the first instance, you must select the measurement items. These items are generated by the information passed by the respondents, through the methods used, and, correlating them with the knowledge of the subject in the literature review. The information generated by the sample is provided through the answer to open questions. In total, 21 respondents, with travel habits, were selected, randomly interviewed through the Focus group and Face-to-face. The two questions (positive and negative) were only asked when the respondent felt difficulty in answering, if not just the first. Each idea was pointed out manually on paper or digital. Respondents took between two and four minutes to answer. Next, results were screened, eliminating common answers.

Validation of the Generated Items—Expert Judgment

According to Churchill [45], this type of evaluation consists of a trial by a certain jury that may offer some knowledge and ideas for the phenomenon. The experts should check and analyze the items provided by the respondents, as well as identify the items that contribute least and the items that contribute most to tourism responsibility from the tourist point of view. On the basis of their nomination, the first variables of tourism responsibility will be understood, which in turn will be used for the research instrument, the questionnaire applied to the target population. For the decision making, three qualified and recognized jurors in the area of tourism were nominated. The experts, after evaluating the generated items, exclude the items they do not consider important and give new considerations or suggestions. In 53 initial items, the first filtering was performed and 22 items were obtained to apply to the questionnaire (See Table 2). The final items obtained to evaluate the question 'What is it for you to be a Responsible Tourist?

**Table 2.** Items selected by the expert panel.

1.  Knowing how to respect the local community
2.  Have environmental responsibility
3.  Do not damage the patrimony
4.  Be compliant with the rules/legislation of the country you visit
5.  Respect the local culture and tradition
6.  Do not throw garbage on the floor
7.  Buying local products/contributing to local trade
8.  Get informed before visiting the site (rules of conduct, religion and customs, appropriate clothing, etc.)
9.  Being civilized
10.  Be patient
11.  Be participatory
12.  Be flexible (know how to adapt)
13.  Be polite
14.  Act consciously
15.  Knowing how to take security measures at your destination
16.  Do not visit forbidden and dangerous places
17.  Knowing how to avoid superfluous consumption (*Souvenirs, amenities,...*)
18.  Vaccination and Travel Insurance
19.  Opting to hire a local tourism agent responsible for making the visits
20.  Acting in a sustainable way (opting for the use of public transport, etc.)
21.  Choose to visit an alternative destination to a mass destination (with many tourists)
22.  Opting to visit a destination that fosters decent and fair conditions and respects the rights of workers

*4.2. Quantitative Study*

Sample 1: Questionnaire

Like the Bassi [51] scale design, the scale proposed in this research was developed and tested in a convenience sample, being evaluated for its reliability and validity through the Zaichkowsky [52] protocol and proposed by Churchill's [45] development procedure of a measuring scale.

Respondents respond to every 22 items using a 5-point Likert-type scale, where 1 = Totally disagree; and 5 = Totally agree [47]. The question: 'What is it for you to be a Responsible Tourist?' and the final items obtained were entered into the online survey platform—Google Forms. Since the objective was to test the relevance of the issues, a non-probabilistic convenience sample was used combined with snow-ball technique. The first sample of 200 respondents was obtained, in which the characteristics of habit and frequency in travelling in tourism are common.

Data Collection: Sample

The questionnaire was applied directly and in person to individuals, as well as on Facebook and LinkedIn social networks, 200 valid answers were obtained (Table 3). Strategically, it was published online through the personal page and in publications of

specific groups linked to the area of tourism, travel, environmental responsibility and social responsibility.

**Table 3.** Respondents' Profile—Sample 1 (*n* = 200)

|  | **N** | **%** |
|---|---|---|
| Gender |  |  |
| Male | 58 | 28.50% |
| Female | 143 | 71.50% |
| Age |  |  |
| 18–25 years | 40 | 20% |
| 26–35 years | 53 | 26.50% |
| 36–45 years | 57 | 28.50% |
| 46–55 years | 29 | 14.50% |
| 56–65 years | 16 | 8% |
| 66–75 years | 4 | 2% |
| Education |  |  |
| Elementary School | 2 | 1% |
| Technical-professional course | 20 | 10% |
| High School | 30 | 15% |
| College education | 149 | 74.50% |
| Travel frequency in tourism per year |  |  |
| At least 1x per year | 82 | 41% |
| 2x per year | 69 | 34.50% |
| 3x per year | 27 | 13.50% |
| 4x or more times per year | 22 | 11% |

*4.3. Step 2—Item Purification*

This second phase of building the scale is the second filtering of the study. At this stage of the scale development, the data collection of sample 1 is submitted to exploratory factor analysis and varimax analysis to reduce the number of items and generate groups, which will be the dimensions of tourism responsibility [47]. The scale purification process is used to reduce the number of items [45]. Based on the study by Fatma et al. [53], to purify the scale the correlations of the items for all statements are examined.

In order to correlate the 22 variables in groups, the correlation matrix was first performed. This matrix shows the correlation between the variables of the study. Only the correlation between the same variables, that is, the crossing between the two is possible to obtain the maximum value of 1. To test whether the correlation between the variables is sufficiently reliable, significance tests are performed. To test significance, the Bartlett sphericity test and the Kaiser–Meyer–Olkin (KMO) test are performed. KMO and Bartlett's are calculated to assess sample suitability [53]. In this study, we observe the KMO of 0.955 which shows that there is correlation between the variables. When analyzing the Bartlett's sphericity test, it is observed a significance of 0.000, what through the level, normally used, of significance of 0.05, evidences that there is correlation between some variables. As the *p*-value (sig) is lower than the significance level (normally used to evaluate 5%), it means that the analysis is adequate. It is concluded that the sample is acceptable for exploratory factor analysis given the sample size and adequacy [52]. Another test is the analysis of communalities. The value of all items was superior to the cutoff point of 0.6 [54].

We also conducted exploratory factor analysis. The total sum of the explained variance is estimated at 76% for three factors, surpassing the minimum value of 60%, confirming the validity of the extraction. The factorial exploratory analysis generated three groups. The rotation method used was that of Varimax with Kaiser normalization. According to Chen and Raab [55], it is usual for researchers to use some of the following criteria to reduce items such as: the variables must have a self-value above 1, and be justified with at least 4% of the total variance; they suggest the elimination of items with values below 0.5 and communalities below 0.3.

Therefore, when performing the Varimax analysis, the variables with coefficients <0.4 were eliminated, allowing the reduction in items that are not important for the study and also taking advantage of the reliability test, through the Cronbach's Alpha test [56]. If the variable's Alpha is inferior to 0.7 is also excluded from the list [57]. The results are:

**Group 1** (Cronbach's Alpha = 0.978) consists of the following 13 variables:

1. Know how to respect the local community
2. Have environmental responsibility
3. Do not damage the patrimony
4. Be compliant with the rules/legislation of the country you visit
5. Respect the local culture and tradition
6. Do not throw the garbage on the floor
8. Inform yourself before visiting the site (rules of conduct, religion and customs, proper clothing, etc.)
9. Being civilized
10. Be patient
12. Be flexible (know how to adapt)
13. Being polite
14. Acting consciously
16. Do not visit forbidden and dangerous places

**Group 2** (Cronbach's Alpha = 0.817) consists of the following four variables:

7. Buying local products/contributing to local trade
11. Being participatory
20. Acting in a sustainable way (opting for the use of public transport, etc.)
22. Opting to visit a destination that fosters decent and fair conditions and respects the rights of workers

**Group 3** (Cronbach's Alpha = 0.767) consists of the following two variables:

15. Know how to take security measures at your destination
18. Vaccination and Travel Insurance

### 4.4. Step 3—Final Validation

The scale for measuring tourist responsibility from the tourist's perspective was developed after conducting the procedures and analyses of the previous sections, however researchers suggest that the scale should be re-analyzed with the purified items using a new sample [45,47]. For the second sample a new scientifically approved measurement scale is introduced to the previous questionnaire. The second questionnaire is then oriented to ensure the validation and reliability of the scale of tourism responsibility [47].

The scientific measurement scale applied in the second questionnaire is that of Tsaur et al. [47], on the tourist-resident cultural conflict, which in its research is measured only for Thailand, and in this study applied to any tourist destination. Like the first questionnaire, respondents should answer, on a Likert scale of 1 to 5, where 1, they totally disagree and 5, they totally agree, but now to two questions: 'What is it for you to be a Responsible Tourist?' and 'In a tourist destination, consider that there is no cultural conflict when'. In order not to be absolutely similar to the first questionnaire, the measurement items to the first question have been introduced in a mixed way so as not to induce equal answers from those who answered the first questionnaire. In this final questionnaire, the respondents started to evaluate 19 purified items from the first questionnaire, and the six items added from the tourist-resident cultural conflict scale [47], with a total of 25 items, in order to verify and validate the final scale of tourism responsibility, through confirmatory factor analysis.

Data from the second sample were collected, according to the first collection and also based on the study by Thomas et al. [46], by a convenience sample of 200 respondents. We followed the same approach of study 1 a used a non-probabilistic convenience sample was used combined with snow-ball technique. Thus, the second questionnaire was also

applied directly to individuals on Facebook and LinkedIn social networks. Table 4 shows the characterization of the respondents.

**Table 4.** Respondents' Profile—Sample 2 (*n* = 200).

|  | **N** | **%** |
|---|---|---|
| Gender |  |  |
| Male | 70 | 34.80% |
| Female | 131 | 65.20% |
| Age comprised |  |  |
| 18–25 years | 29 | 14.40% |
| 26–35 years | 64 | 31.80% |
| 36–45 years | 55 | 28.40% |
| 46–55 years | 41 | 20.40% |
| 56–65 years | 9 | 4.50% |
| 66–75 years | 3 | 1.50% |
| Education |  |  |
| Elementary School | 2 | 1% |
| Technical-professional course | 29 | 14.40% |
| High School | 38 | 18.90% |
| College education | 132 | 65.70% |
| Travel frequency in tourism per year |  |  |
| At least 1x per year | 78 | 38.80% |
| 2x per year | 61 | 30.30% |
| 3x per year | 30 | 14.90% |
| 4x or more times per year | 32 | 15.90% |

The second questionnaire was applied to another convenience sample, obtaining 200 valid responses. In order to proceed with the verification of the accuracy of the study it is necessary to perform confirmatory factor analysis. Confirmatory factorial analysis is applied to verify the scale generated in the previous step [6], and to evaluate the quality and significance of the measure, and to verify the discriminant validity from similar scales [55]. The results of the confirmatory analysis using Structural Equation Modeling performed through the SmartPLS 3 software [57], showing that tourism responsibility is influenced by only two dimensions.

The scale of tourist responsibility, when applied to confirmatory factorial analysis with the SmartPLS program, is explained by the type II multidimensional reflexive-formative equation model, third order index in the model construction [58,59]. It was found that after all, the previously proven scientific scale was not included in the confirmatory analysis and that only 11 variables have an impact in 2 dimensions, considered as second order constructs [59] and, in turn, influence a third order, the responsibility of the tourist. The second and third order constructions are formative because they influence the 11 items used to measure them, and which, in turn, are considered as first order reflective constructions. The explanation for the scale being determined as a model of a multi-dimensional reflexive and formative equation of type II is proven through the measurement of the variation-inflation factors (VIF) and the values of the coefficients (*t*-Values). It is verified that all VIFs are lower than 0.5, confirming the doctrine [59], and, in parallel, that the values of the coefficients between the first and second order are significantly high, i.e., greater than 1 [59]. In order to evaluate also the reflective constructs (measurable variables), the data analysis of the load of the items, known as 'Loading Item', the composite reliability or Composite Reliability (CR), the mean extracted variation known as Average Variance Extracted (AVE) [59] and the Cronbach's Alpha was performed. Observing Table 4 below, it is verified that the data are within the adequate values.

### 4.4.1. Convergent Validity

For Pasquali [60], convergent validation is defined as the significant relationship between two measures or constructs that are theoretically related. The convergent validation between the dimensions (constructs) is evaluated by calculating the Mean Extracted Variation (AVE) and the Composite Reliability (CR) for all constructs [60,61]. The convergent validation of the scale is confirmed by the analysis of the mean extracted variation (AVE) for each dimension was at least 0.5, the value of at least 0.7 for the composite reliability (CR) [62] and the load of each measurement item is above the threshold of 0.5 and the significant value of, $p < 0.001$ [62]. The presented values of AVE of both dimensions are higher than 0.5, in the civic responsibility dimension of 0.7 and in the philanthropic responsibility dimension of 0.686, CR of 0.949 and 0.867, respectively. As for the load of each variable, all 11 variables have values higher than 0.5. With respect to Cronbach's Alpha that tests the reliability of each dimension, both dimensions present values higher than 0.7. The results demonstrate the validity of convergence of the built scale.

### 4.4.2. Divergent Validity

Hair et al. [62] considers that the divergent validity of a construction measure should be empirically unique and represent phenomena of interest that another model of structural equation does not present. For this purpose, the standard used to assess the discriminant validity was proposed by Fornell and Larcker [63] and suggests that the square root value of the mean extracted variation (AVE) of each construct should be greater than the value of the correlations between the constructs [62].

By looking at Table 5 it can be seen that the constructs, designated by the Civic Responsibility Dimension (CRD) and Philanthropic Responsibility Dimension (PRD) dimensions have divergent validity since the correlation between the dimensions have values lower than the AVE value squared each one (Table 6), clarifying that their variables are better explained by their respective constructs and not the other way around.

**Table 5.** Measuring scale of tourist responsibility.

| Dimension | Definition | Variables | Item Loading | *t*-Value | CR | AVE | Cronbach Alpha |
|---|---|---|---|---|---|---|---|
| Civic Responsibility Dimension (CRD) | Tourist who avoids damages and is responsible for doing what is right, right. | 4. know how to take security measures at your destination | 0.786 | 18.403 | 0.949 | 0.7 | 0.938 |
| | | 5. be compliant with the regulation or legislation of the country you visit | 0.874 | 28.811 | | | |
| | | 6. Act consciously | 0.900 | 41.276 | | | |
| | | 7. be flexible (know how to adapt) | 0.869 | 31.885 | | | |
| | | 11. get vaccinated and choose to take out travel insurance | 0.686 | 12.661 | | | |
| | | Inform yourself before visiting the site | 0.809 | 19.228 | | | |
| | | 14. Do not throw garbage on the floor | 0.869 | 18.677 | | | |
| | | 19. To be civilized | 0.879 | 30.519 | | | |
| Philanthropic Responsibility Dimension (PRD) | Tourist who contributes resources and improves the quality of life of the community he visits. | 1. have environmental responsibility | 0.885 | 54.695 | 0.867 | 0.686 | 0.774 |
| | | 2. be patient | 0.781 | 19.597 | | | |
| | | 3. Buying local products/contributing to local trade | 0.814 | 24.507 | | | |

**Table 6.** Divergent validity.

| Buildings | AVE | DRC–DRF Correlation | AVE |
|---|---|---|---|
| CRD | 0.700 | | 0.840 |
| | | 0.742 | |
| PRD | 0.686 | | 0.830 |

### 4.4.3. Nomological Validation

According to Bagozzi and Yi [57], the nomological validity is the degree to which a main building behaves within a system of related constructions. To test the nomological validity the structural model of the equation composed by Figure 3; Figure 4 were elaborated.

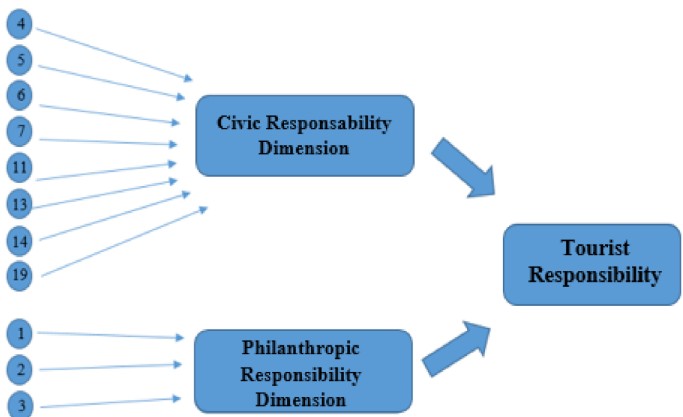

**Figure 3.** Confirmatory Factorial Analysis: Model of Structural Equation Type II.

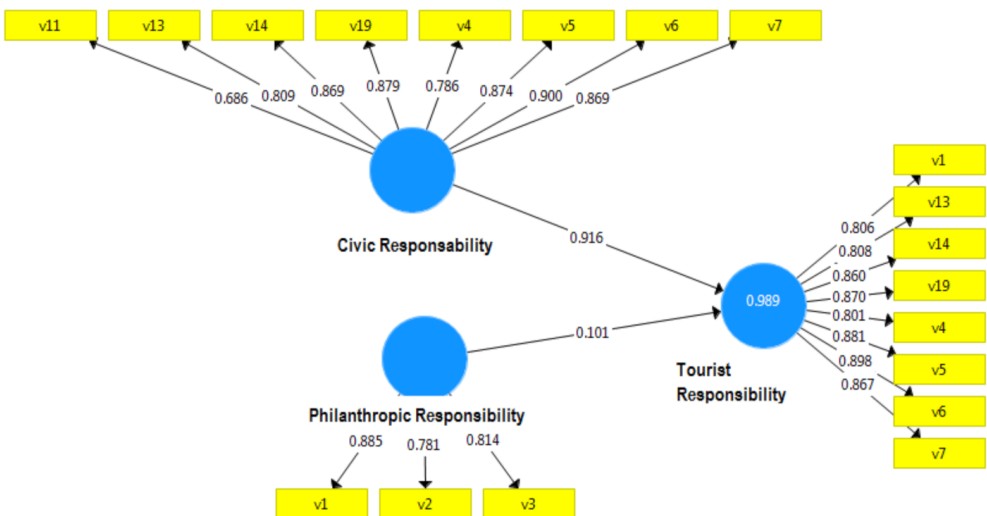

**Figure 4.** Partial Least Squares outputs of the model.

## 5. Discussion

In the first two methodological studies, qualitative methods were used to generate the scale items. Subsequently, quantitative studies were conducted using convenience samples to conduct the research. Both samples proved to be significant for conducting factor analyses. To test the three dimensions generated, the science of developing a measurement scale implies reviewing the exploratory factorial analysis with the 19 items, three less than in the first study. It is necessary to assess the validity of the constructs by introducing another similar scale. A new quantitative methodological study is performed to obtain a new convenience sample, to run confirmatory factor analysis. The confirmatory factor

analysis reduced the scale to 11 items distributed by two dimensions: civic responsibility and philanthropic responsibility.

These two dimensions confirm the theory of responsible tourism: contributing to and improving the quality of life of communities, cultures, environments and local economies, and minimizing negative impacts on territories [17]. The first dimension, civic responsibility, refers to tourism practices related to respect, awareness, and interaction with the environmental, cultural, and social characteristics of the destination visited, preserving the local cultural and natural environment [35]. For Krippendorf [22], responsible tourists are willing to invest their time and resources in seeking information before and during their trip in order to experience the local context in a conscious and ethical manner. According to Martins [64], it is possible to relate civic being with being responsible. For the author, being a responsible citizen is about the commitment and responsibility of a person in a community. In the context of action, commitment, and responsibility must prevail. Commitment can thus be related to civic-mindedness, in that it can be considered an obligation and personal commitment in relation to a decision or plan, which refers to responsibility.

This dimension corresponds to the dimension of civic responsibility which is reflected in the tourist code of ethics [24], thus admitting that one of the dimensions of civic responsibility is the ethical attitude of the tourist. It should be noted that ethics and responsibility are terms that often converge in the tourism literature and have been used interchangeably by some researchers [17,35]. Items such as buying local products, being participative, acting in a sustainable way, and choosing to visit a destination that fosters decent conditions for workers, are latent variables in the criteria of sustainable practice in destinations, understood by tourism sustainability, which in turn is intrinsically linked to the practice of responsible tourism, and is often considered as the practical application of the concept of sustainability [25]. The philanthropic dimension comes from the concept of philanthropy of travelers, which for Goodwin [17] is considered the tourist's responsibility to give, to the places or the communities, money, or time. Through the Traveler's Philanthropy Manual created by the American Center for Responsible Travel [65], the civic attitude of tourists, as well as tour operators, have contributed to the improvement of the management of the destinations visited, and the promotion of the well-being of local populations. For Honey [65], the philanthropic dimension of tourists positively favors universal citizenship through the learning and pleasure that travelers offer for community projects and in the conservation of tourist destinations. The philanthropic nature of tourists will be the next step in the continuous development of responsible tourism, as it allows the tourist to better understand communities, allows the best flow of income, helping long-term projects and contributing to knowledge in supporting the three fundamental pillars of sustainability: economic, social, and environmental well-being.

For Burrai et al. [66], the ideology of responsible tourism becomes evident if we focus on moral values and sustainability, the defense of local interests and human rights [66]. The remaining variables are admitted to the second dimension created, the dimension of philanthropic responsibility, consisting of variables such as the notion of the tourist's environmental responsibility, his contribution to local commerce, and the virtue of having the capacity to be a patient tourist. These determinants are justified through the notion of philanthropic responsibility of the tourist, related to his kind and responsible practice in the territories, that for Goodwin [17] these actions create better places for people to live and visit. These aspirations refer to responsible tourism as a way for tourists to improve their livelihoods and to know how to maintain, protect, and improve the places where these livelihoods occur [66]. Goodwin [17] further defines responsible tourism as a social movement, and describes it as an intentional effort by groups of people who know how to share common principles and approaches, resulting in a shared sense of direction.

The second dimension, referring to the philanthropic responsibility of the tourist, emphasizes practices related to interaction with the local population, showing that a responsible tourist tends to be an active individual, open to all kinds of social experiences [35]. Leslie [15] also specifies that responsible tourism is concerned with people, the

environment, values, and culture, in addition to being able to reduce negative impacts by improving working conditions, enabling community involvement, promoting cultural heritage, and protecting the environment.

Responsible tourism is a matter of respect, awareness, and education through local identity and interaction with the locals and with whom tourists can share a common sense of reciprocity and involvement. The theory that tourism responsibility from the tourist perspective is measured through the dimension of civic responsibility and the dimension of philanthropic responsibility is in line with the proposal defended by several authors [22,35]. Like responsible tourism, the concept of tourism sustainability, which is a prerequisite for sustainable tourism, suggests balancing the economic, socio-cultural, and ecological dimensions for the appropriate development of tourism in the territories [66]. Accordingly, Buckley [67] concludes that the four most popular areas of sustainable tourism are ecotourism, responsible tourism, community-based tourism, and conservation tourism. Based on the UNWTO [24], the variables presented in these dimension into the extent to which they meet the analysis between the ecological, the socio-cultural and the economic. The World Tourism Organization considers that these three levels of tourism sustainability guarantee the environmental development, which by analyzing from the tourist point of view, through its sustainable action in the territory is possible to be achieved. The choice of the tourist for a decent tourist destination and the fact that it is participatory, contributes to the autonomy of local communities, preserving their culture, identity and values, guaranteeing the future generation, and economically, the tourist contributing to the purchase of local products, allows economic development and proper management of resources.

## 6. Conclusions

### 6.1. Theoretical Implications

Considering the central objective of this research, the results allow establishing a scale to measure tourists' responsibility composed of two dimensions: civic responsibility dimension and philanthropic responsibility dimension. Having the results been robust regarding the scale's reliability, this paper contributes to the knowledge in the area of tourism sustainability with a tool that allows other research studies to build on this scale to create conceptual models. The creation of the scale and the identification of these two dimensions result in the novelty of the study in advancing previous knowledge by adding a quantitative approach to the tourist responsibility measurement [5,6].

Although the difficulty in distinguishing between responsible tourism and sustainable tourism is understandable, responsible tourism should be considered as an agent for sustainable development in a destination [16], because sustainable tourism is more related to sustainability awareness, while responsible tourism is the most practical form of action for sustainable tourism. Responsible tourism generates sustainable tourism being achieved by taking responsibility for the consequences of the actions taken [18]. Thus, responsible tourism is the behavioral trait [15,16] and from this characteristic comes responsible behavior in tourism.

Some disparities are evident between tourists' perceptions of responsible attitudes and their actual behavior in practice [33]. Wheeler [10] admits that tourists often want to travel without feeling guilty about what they do, that is, without having to retract their behavior or pleasure. For the author, responsible tourism is a pleasant but dangerously superficial escape route for those who are unable to accept their own destructive contribution to the negative impacts that can generate their tourist practice.

There may be several motivations for tourists to compromise their responsible behavior in the territories, one of which is the difficulty in finding information on responsible tourism products and services, on the appropriate conduct to have and also on the specific nature of a place [2].

More recently, several authors [68–71] have pointed to a change in the responsible behavior of tourists as a result of the pandemic caused by COVID-19, namely different

travel preparation [69], changes in consumption [68], increased preference for domestic travel [70], and the search for more innovative tourist experiences [71]. All of these findings share a common dimension: a desire for more sustainable and responsible practices.

### 6.2. Managerial Implications

Destination managers, known as DMO, are responsible for ensuring that the growth of the activity is well managed, that the benefits are maximized and that any negative externalities should be minimized. This requires a continuous planning and management process, measured over the long term. It is believed that the most effective contribution to bringing responsibility to tourism is through direct action by policy makers [15].

It is necessary to raise the awareness of tourism agents and tourists through a good communication policy, to lead to the understanding of the effects that their behaviors generate and how they can be mitigated to help reduce the negative climatic and environmental, economic, socio-economic, and cultural consequences of the development of tourism activity in the territories [40].

It is accepted that it is necessary to identify and provide information and ethical actions to increase the awareness and sense of responsibility of tourists and tourism agents [2]. Accordingly, it is fundamental to develop incentive policies for the action of responsible tourism activity.

In order to strengthen tourist practices, it is advised that educational programs should be targeted to make tourists aware of concerns that might compromise the success of tourism in a particular destination [40]. Governments should continue to disclose financial incentives, through subsidies or tax benefits, to tourism companies that provide and disclose responsible tourism activities to tourists [7]. Governments should enact mandatory ecological and responsible certification for tourism companies, thus encouraging and making the tourist responsible and ethical practice, contributing to the reduction in their ecological footprint. It is also suggested that information of an essentially cultural-nature-based in a given territory and on the carrying capacity of the territory and the period when tourist flows are greatest should be created and made available free of charge, thus avoiding the over-tourism of regions, for example when buying a tourist package in travel agencies or during air travel. Virtual companies and other technological applications provide trends to encourage this type of tourism, such as the Pinterest application on trends to travel responsibly, including train travel; the use of zero waste travel essentials; the concept of Stay-vacations, in traveling close to home; visit environmentally friendly cities; practice agro-tourism and ecotourism; exchange hotel stays for ecological structures; share travel stories to advise and warn future travelers. These and other practices can contribute to increase the individual's awareness in choosing one destination over another, as well as helping to decentralize tourism activity in the regions, a latent concern in the sustainable management of destinations and fundamental for the increase in responsible tourism on a global scale.

### 6.3. Limitations and Future Research

The contribution to scientific tourism is a constant for science. The trend towards responsible tourism is identified in the most recent research. The dimensions of tourism responsibility, in general, cover all areas, economic, social, cultural, environmental, and even political, and it is certain that all stakeholders in tourism activity are responsible for the actions they trigger.

There are some limitations to this investigation. The first is the selection of articles to deal with the subject. A significant number of researches are clearly addressing the notion of tourist responsibility, however, there are not many studies on the specificity of tourist responsibility itself. Another benchmark refers to the application of questionnaires, essentially online. It is known that the conduct of online questionnaires is always dubious in the veracity of the answers obtained. Additionally, it is important to mention that, although the methodology used is tested in scientifically designed software, it is fundamental to

understand that the behavior of the tourist, is multifaceted and complex [34], that is, it is changeable from individual to individual and changeable over time and through the environment in which it is inserted. In order for the scale of tourist responsibility to be real, it is also necessary to study the authentic performance of the tourist's behavior and compare it to the idea he has about how to act responsibly in destinations. There is also another limitation in the study. This research does not delimit a destination, territory or country; it is a study produced with a general conformation to the travelling population, which may not guarantee that it can be used in all cultures and tourist contexts. Thus, it is suggested that in future research, this study may extend to a context of a specific region using the attributes and dimensions examined, applied to a more representative sample. Future researchers are recommended to approach the present object of study more extensively, exploring the phenomenon from the perspective of the stakeholders, and it may contribute to the creation of one or more dimensions of tourism responsibility from the perspective of the other tourism stakeholders.

One final limitation is the use of a non-probabilistic convenience sample in both quantitative studies, which can limit the generalization of the results.

**Author Contributions:** Conceptualization, Á.D. and I.A.; methodology, Á.D. and I.A.; validation, L.P., R.L.d.C. and N.A.; formal analysis, I.A.; investigation, Á.D. and I.A.; resources, N.A.; writing—original draft preparation, I.A.; writing—review and editing, L.P. and R.L.d.C. All authors have read and agreed to the published version of the manuscript.

**Funding:** No funding to report.

**Data Availability Statement:** Data will be available upon request.

**Conflicts of Interest:** Nothing to declare.

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
