# Peer review of "A Measure of Tourist Responsibility"

_sustainability, doi:10.3390/su13063351_

Round 1
Reviewer 1 Report
Dear Authors,
After read your work I have this considerations to do:
1-Abstract: Presented correctly and clearly. It understands if the objective of the authors. I would like more precise information about the methods used and their sample to be introduced. In the methodology they should introduce how the data were treated and with which software. In the sample they should characterize the number of participants and population universe.
2-Introduction: In this section the authors focused much on explaining how the study is done and its methodology when they should make the general framework of the theme, enunciate research questions and objectives. The methodology part has its own section and is not worth repeating. In the introduction you should also focus your efforts on the research GAP. You should also state how the article will be divided and contributions from each section to the final work.
3-Methodology: Very complete and robust. All scientific steps are explained. I draw the authors' attention to errors in section numbering. Please review.
Other issues:
Section 3 is very complete but it is very large. It needs better organisation and can be subdivided between Methodology and Results. When the authors present results from the application of the scale, these data do not need to be in the methodology as they are already results from the application/test of the scale. This section cannot be so large because it gets disorganized. Authors should create a results section where they insert the statistical tests obtained in scale tests.
Important Note/My personal Opinion:
The authors have chosen to present all the results in a phased manner and very well, however, they present the results of the exploratory factorial analysis and then present the confirmatory model but in this model they do not present results.
However, if they present results in everything, they can also present the PLS estimation results here so that readers can see the scale working already with preliminary results.
What I mean by this is: either they don't present the model in this article and do it in another article or, if they want to present everything then they should also show the estimated model.
In my personal opinion this article could be divided into two articles. The proosta of the scale and the exploratory factorial analysis . And another article with an empirical study using this scale and presenting a confirmatory factor analysis with hypothesis tests. But, I leave it to the authors to consider this final decision, which I will accept whatever it is your final decision in next revisions round.
Discussion and Conclusions: Very very good. If you change the final paper followed my opinions please adjust this sections.
Final Note:
High quality article. Very complete with hard work done by the authors.
After small corrections it will deserve publication in this important Journal.
I wish the authors good work and much health.
My best Regards
Author Response
Dear Reviewer,
Thank you very much for your important comments. We implemented all of them as you can see below:
1-Abstract: Presented correctly and clearly. It understands if the objective of the authors. I would like more precise information about the methods used and their sample to be introduced. In the methodology they should introduce how the data were treated and with which software. In the sample they should characterize the number of participants and population universe.
R: We agree with the reviewer. In fact, the detail about the sampling approach is missing. As such, we now introduce an explanation with regard the use of a non-probabilistic convenience sample combined with snow-ball technique. Since it is not representative of the overall population universe we think there is no need to include it. However, we recognize this as a limitation in the end of the article. Please see these changes in Section 4.2.1. second paragraph, 4.4. third paragraph and 6.3 third paragraph.
2-Introduction: In this section the authors focused much on explaining how the study is done and its methodology when they should make the general framework of the theme, enunciate research questions and objectives. The methodology part has its own section and is not worth repeating. In the introduction you should also focus your efforts on the research GAP. You should also state how the article will be divided and contributions from each section to the final work.
R: Thank you for pointing this out. Although we presented a framework supporting the lack of a measure similar to ours, we understood that it was not sufficiently clear. As such, we re-write the second paragraph of the introduction for present more clearly the research gap. We also included the article structure and the contribution of each section. Please see last paragraph of the introduction.
3-Methodology: Very complete and robust. All scientific steps are explained. I draw the authors' attention to errors in section numbering. Please review.
R: Thank you for pointing this out. The sections numbers are now revised and changed for the adequate sequence.
Other issues:
Section 3 is very complete but it is very large. It needs better organisation and can be subdivided between Methodology and Results. When the authors present results from the application of the scale, these data do not need to be in the methodology as they are already results from the application/test of the scale. This section cannot be so large because it gets disorganized. Authors should create a results section where they insert the statistical tests obtained in scale tests.
R: Thank you for pointing the suggestion. The section 3 is now divided in section 3 Methodology and section 4. Results.
Important Note/My personal Opinion:
The authors have chosen to present all the results in a phased manner and very well, however, they present the results of the exploratory factorial analysis and then present the confirmatory model but in this model they do not present results.
However, if they present results in everything, they can also present the PLS estimation results here so that readers can see the scale working already with preliminary results.
What I mean by this is: either they don't present the model in this article and do it in another article or, if they want to present everything then they should also show the estimated model.
In my personal opinion this article could be divided into two articles. The proosta of the scale and the exploratory factorial analysis . And another article with an empirical study using this scale and presenting a confirmatory factor analysis with hypothesis tests. But, I leave it to the authors to consider this final decision, which I will accept whatever it is your final decision in next revisions round.
R: Thank you for this suggestion. In fact, we have material to produce a second article, since a part of the article was cut off to reduce the initial size of this study. Accordingly, we will start composing the second article based on your valuable suggestion (many thanks). However, we also believe that the model estimation could illustrate the relations between the dimensions and the final construct. As such, we included Figure 5 to present this results.
Discussion and Conclusions: Very very good. If you change the final paper followed my opinions please adjust this sections.
R: Thank you for pointing this out. The sections were renumbered considering the creation of the section ‘results’.
Final Note:
High quality article. Very complete with hard work done by the authors.
After small corrections it will deserve publication in this important Journal.
I wish the authors good work and much health.
My best Regards
R: Thank you for the contribution and for these supportive comments.

Reviewer 2 Report
In the manuscript, the authors presented an important point, although the manuscript has some drawbacks:
1. Intruduction - please write about sustainability. The manuscript was sent to Sustainability.
2. Conclusions - please refer to the COVID-19 pandemic and the presented problem in the manuscript. What about COVID and tourism? Suggested publications:
• Kitamura, Y .; Karkour, S .; Ichisugi, Y .; Itsubo, N. Evaluation of the Economic, Environmental, and Social Impacts of the COVID-19 Pandemic on the Japanese Tourism Industry. Sustainability 2020, 12, 10302. https://doi.org/10.3390/su122410302
• Roman, M .; Niedziółka, A .; Krasnodębski, A. Respondents' Involvement in Tourist Activities at the Time of the COVID-19 Pandemic. Sustainability 2020, 12, 9610. https://doi.org/10.3390/su12229610
• Sung, Y.-A; Kim, K.-W .; Kwon, H.-J. Big Data Analysis of Korean Travelers' Behavior in the Post-COVID-19 Era. Sustainability 2021, 13, 310. https://doi.org/10.3390/su13010310
In your conclusions, please also answer the following questions:
• what are the research gaps?
• what is new to this manuscript?
Author Response
Dear Reviewer,
Thank you very much for your important comments. We implemented all of them as you can see below:
In the manuscript, the authors presented an important point, although the manuscript has some drawbacks:
- Intruduction- please write about sustainability. The manuscript was sent to Sustainability.
R: We agree with the reviewer. We now incorporated in the narrative this important concept. Please see paragraphs 1 and 3 of the introduction.
- Conclusions- please refer to the COVID-19 pandemic and the presented problem in the manuscript. What about COVID and tourism? Suggested publications:
- Kitamura, Y .; Karkour, S .; Ichisugi, Y .; Itsubo, N. Evaluation of the Economic, Environmental, and Social Impacts of the COVID-19 Pandemic on the Japanese Tourism Industry. Sustainability 2020, 12, 10302. https://doi.org/10.3390/su122410302
• Roman, M .; Niedziółka, A .; Krasnodębski, A. Respondents' Involvement in Tourist Activities at the Time of the COVID-19 Pandemic. Sustainability 2020, 12, 9610. https://doi.org/10.3390/su12229610
• Sung, Y.-A; Kim, K.-W .; Kwon, H.-J. Big Data Analysis of Korean Travelers' Behavior in the Post-COVID-19 Era. Sustainability 2021, 13, 310. https://doi.org/10.3390/su13010310
R: Thank you for the suggestion and for the references. We now included this discussion and respective references in the conclusions as suggested. Please see 5th paragraph of the theoretical conclusions.
In your conclusions, please also answer the following questions:
• what are the research gaps?
• what is new to this manuscript?
R: We agree with the reviewer. We now added additional elements to the conclusions by highlighting the study contribution and novelty. Please see section 6.1. first paragraph.

Reviewer 3 Report
The authors undoubtedly chose a very current topic for the article, as well as a suitable research method. However, I recommend the following content, methodological and formal adjustments.
Content adjustments:
The theoretical part presents the view that "Chan and Xin [12] also consider responsible tourism as a positive alternative to mass tourism and capable of replacing it. However, it would be appropriate to state that responsibility in tourism (or responsible tourism) can have a significant role to play and (albeit relatively few so far) in mass tourism. For inspiration e.g. David B. Weaver, Organic, incremental and induced paths to sustainable mass tourism convergence, Tourism Management, Volume 33, Issue 5, 2012, Pages 1030-1037,
https://doi.org/10.1016/j.tourman.2011.08.011.
Formal adjustments:
"Source: The authors" is not listed for tables and figures
Figure. 1 - the figure does not show all the tourism stakeholders that are listed in the text of the figure
Figure 3. Confirmatory Factorial Analysis: Model of Structural Equation Type II. Source: Own elaboration.
Source: The authors
It is verified that all FEVs are lower than 0.5, confirming the doctrine (Hair et al., 2011) - the [1] citation style should be used
De several perspectives- The several perspectives
The United Nations [22], in 1999, projected the guiding principles for the participants in tourism (governments, tourism industry and communities), through the realization of the global ethical code for tourism - rightly be the UN WTO
Methodology adjustments:
The introductory part of methodology needs to be adjusted, it cannot be described as how the research could be carried out. It is necessary to write more clearly which method and why it was used (eg in the section "It is known that item generation is composed of two methods: deductive and inductive. The deductive methods are based on the items created through the knowledge of the literature review, and also based on pre-existing scales, on the subject studied.The inductive methods are the qualitative information obtained by the public opinions, usually by focus group and interviews". The form of the description in the methodology should also be shorter.
Author Response
Dear Reviewer,
Thank you very much for your important comments. We implemented all of them as you can see below:
The authors undoubtedly chose a very current topic for the article, as well as a suitable research method. However, I recommend the following content, methodological and formal adjustments.
R: Thank you for the contribution and for these supportive comments.
Content adjustments:
The theoretical part presents the view that "Chan and Xin [12] also consider responsible tourism as a positive alternative to mass tourism and capable of replacing it. However, it would be appropriate to state that responsibility in tourism (or responsible tourism) can have a significant role to play and (albeit relatively few so far) in mass tourism. For inspiration e.g. David B. Weaver, Organic, incremental and induced paths to sustainable mass tourism convergence, Tourism Management, Volume 33, Issue 5, 2012, Pages 1030-1037,
https://doi.org/10.1016/j.tourman.2011.08.011.
R: Thank you for this important suggestion and for the reference, which we incorporated in the list of references. The sentence was changed to “Responsible tourism can be considered as a positive alternative to mass tourism and capable of replacing it [12], but it can also play a significant role on mass tourism itself”
Formal adjustments:
"Source: The authors" is not listed for tables and figures
R: We agree with the reviewer. This source is eliminated to avoid read confusion.
Figure. 1 - the figure does not show all the tourism stakeholders that are listed in the text of the figure
R: We agree with the reviewer. Figure 1 has been changed to integrate all the mentioned stakeholders.
Figure 3. Confirmatory Factorial Analysis: Model of Structural Equation Type II. Source: Own elaboration.
Source: The authors
R: Thank you for pointing this out. This source is eliminated to avoid read confusion.
It is verified that all FEVs are lower than 0.5, confirming the doctrine (Hair et al., 2011) - the [1] citation style should be used
R: Thank you for pointing this out. The reference is now changed to the journal format.
De several perspectives- The several perspectives
R: It is indeed incorrect. The word was changed accordingly.
The United Nations [22], in 1999, projected the guiding principles for the participants in tourism (governments, tourism industry and communities), through the realization of the global ethical code for tourism - rightly be the UN WTO
R: We agree with the reviewer. We changed the reference to UNWTO.
Methodology adjustments:
The introductory part of methodology needs to be adjusted, it cannot be described as how the research could be carried out. It is necessary to write more clearly which method and why it was used (eg in the section "It is known that item generation is composed of two methods: deductive and inductive. The deductive methods are based on the items created through the knowledge of the literature review, and also based on pre-existing scales, on the subject studied.The inductive methods are the qualitative information obtained by the public opinions, usually by focus group and interviews". The form of the description in the methodology should also be shorter.
R: We agree with the reviewer. We revised the document, and divided the section 3 in section 3 and section 4, presenting just the methodological approach and the results, respectively. We also understood the example you gave and eliminated superfluous content.

Round 2
Reviewer 2 Report
Accept in present form. Good luck!